# Topological quantum computation using analog gravitational holonomy and time dilation

**Emil Génetay Johansen⋆ and Tapio Simula**

Optical Sciences Centre, Swinburne University of Technology

⋆ egenetayjohansen@swin.edu.au

## Abstract

Non-universal topological quantum computation models, such as the Majorana fermion-based Ising anyon model, have to be supplemented with an additional non-topological noisy gate in order to achieve universality. Here we endeavour to remedy this using an Einstein–Cartan analog gravity picture of scalar fields. Specifically, we show that the analog gravity picture enables unitary transformations to be realized in two distinct ways: (i) via space-time holonomy and (ii) as gravitational time dilation. The non-abelian geometric phases are enabled by gravitational interactions, which are mediated by the spin-connection. We analytically compute its matrix elements as a function of the scalar field density distribution. This density can be regarded as the gravitating distribution of matter in an analog universe. We show via explicit calculations that there exists an infinite set of asymptotically flat analog gravitational fields, each of which implements a unique unitary transformation, that render the interactions topological. We emphasise the generality of this result by asserting that such gravitational gates could potentially be implemented in a broad range of real systems modeled by scalar field with an acoustic metric.

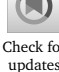
# 1 Introduction

Low-dimensional physical systems have proven to provide a fertile ground on which the interplay between gravity and quantum field theory can be studied. In particular, a $(2+1)$-dimensional gravity has been studied intensively since Witten discovered a dualism which connects it to $SO(1,2)_k$ Chern-Simons theory, where the level $k$ on one side relates to the cosmological constant on the other [1]. This suggests that a (2+1)-dimensional gravity is a topological field theory implying that interactions are only manifest as non-local effects. Theories of this type can be simulated in superfluid condensates through *analog gravity* [2–4]. The notion of analog gravity was first introduced in 1981 by Unruh and has been considered in numerous contexts such as black hole physics [4–7], inflationary physics [8–10], rotating universes [11] and cosmic strings [12,13]. Within this framework, the condensate itself constitute the fundamental substance of space-time in which the quasi-particle fields $\phi$ are embedded and the speed of sound plays the role of the speed of light, thus enforcing a causal structure on the space. Consequently, the fields are subject to an effective metric $g_{\mu\nu}$ implying that the dynamics is governed by the equations of motion $\partial_\mu(\sqrt{-g}g^{\mu\nu}\partial_\nu\phi) = 0$ [4].

In addition to gravity, it has also been shown that electromagnetic field theory [14, 15] as well as quantum field theory [16–21] can be simulated in superfluids through the generation of quantum turbulence [22–24]. Hence, we may refer to a model of this kind as a *superfluid universe*. Here the analog space-time may exhibit curvature via the non-vanishing density gradient and in addition torsion in the presence of vortices. The core objective of our work is to address the question whether there exists matter distributions that give rise to topological geometric phases. We show, by explicit calculations, the existence of such solutions corresponding to asymptotically flat splace-times and illustrate that there are in fact two distinct topological effects corresponding to (i) space-time holonomy and (ii ) gravitational time dilation. Topological quantum computers [25–27] based on non-universal gate sets, such as Majorana zero mode (MZM) models [28–33] or bosonic models based on discrete gauge theory [34–37], are not capable of fault-tolerant universal quantum computation by braiding alone. Therefore, in in order to make up for this deficiency, additional features such as non-topological measurement-based protocols [38, 39] have to be adopted. This motivates our searching alternative methods to supplement such non-universal topological quantum computation models.

Considering an analog gravity picture, it seems natural to contemplate exploiting all space-time symmetries to harness unitary transformations to serve as a basis for quantum computation. Within general relativity [40], parallel transporting a vector (or spinor) around a closed loop generally yields a discrepancy in its angle with respect to the initial state. Hence, by coupling a spinor to gravity the holonomy group of the underlying connection should, given a non-flat space-time, generally be non-trivial. Furthermore, this interaction may be made topological giving rise to an effect akin to that in the Aharonov–Bohm experiment [41, 42]. To explore the possibility of exploiting the analog gravity effects, we employ the Einstein–Cartan tetrad formalism [43, 44] and set up a locally flat tangent space at each space-time point, which allows us to work in a local coordinate basis. The Poincaré group constitutes the group of Minkowski space-time isometries, thus leaving the local tangent space invariant, and consequently, this group serves as a gauge group in the analog gravity picture. Note that the gauge group in the conventional hydrodynamic picture is encoded in the gauge group of the gravitational description. However, the gravitational picture reveals that additional topological gauge transformations are embedded within the quantum pressure term of the Madelung transformation [45], and these can be unlocked by carefully choosing the scalar field density. In doing so, we have an access to the full Poincaré group $ISO^+(1,2)$, ready to be deployed for computation. Indeed, breaking the uniformity of the condensate density furnishes the system

with additional degrees of freedom required to achieve computational universality. We construct explicitly a non-abelian energy-momentum flux tube by curving the space-time within a finite region of space, outside which the curvature vanishes. This induces an arbitrary Pauli-$Z$ transformation when encircled by a quasi-particle, as illustrated in Fig. 1(a). We also show that a Pauli-$iX$ transformation can be implemented by simply creating a local over or under density in the manifold in which the quasi-particle sits for some finite time, as illustrated in Fig. 1(b). We note that gravitational analogs of the Aharonov–Bohm effect have previously been considered e.g. in [46–50], albeit not as a basis for computation.

## 2 Superfluid universe picture

For the sake of concreteness, consider a superfluid condensate of bosons governed by a macroscopic wave function $\Psi(r)$, which solves the Gross–Pitaevskii equation [51]. Applying the Madelung transformation $\Psi(r) = |\Psi(r)|e^{i\theta(r)}$, to separate the amplitude of $\Psi$ and its phase $\theta$, the kinetic energy term in the Gross–Pitaevskii energy functional may be split into two terms pertaining to the electromagnetic (phase gradient) and gravitational (amplitude gradient) contributions according to [15]

$$\mathcal{E} = \frac{\hbar^2}{2m} \int dr^3 |\Psi|^2 \left[ |\nabla\theta|^2 + \left( \frac{\nabla|\Psi|}{|\Psi|} \right)^2 \right], \tag{1}$$

where the phase gradient field gives rise to vortex electrodynamics and the amplitude gradient yields a gravitational field in regions of non-zero quantum pressure. It is commonplace to approximate the condensate density to be constant so that the second term vanishes resulting in electrodynamics in flat space-time. In contrast, here we focus on the second term that yields gravity. Allowing for a variable density profile the quasi-particles experience the system as an effectively curved space-time. The quasi-particle excitations of the system are determined by the Bogoliubov–de Gennes (BdG) theory [52], which reveals that the quasi-particle modes are 2-spinor objects $(u_q, v_q)^T$, where $u_q$ and $v_q$ may be associated with particle and anti-particle modes, respectively.

In conventional quantum field theory the space-time which the particle fields inhabit is assumed to be flat. However, generalizations to arbitrary manifolds is possible by the introduction of appropriate connections. Any smooth manifold is locally flat so by attaching to each point $p$ a tangent space $T(p)$, the physics is locally subjected to a Minkowski metric $\eta_{\mu\nu}$. Thus, by the introduction of a connection that patches together two separate tangent spaces $T(p)$ and $T(p')$, where $p'$ is in the vicinity of $p$, field theory in curved space-time can be formulated in terms of field theory in flat Minkowski space-time [53, 54]. In the next section we shall outline this procedure in further detail.

## 3 Einstein–Cartan gravity in a superfluid

Notwithstanding the apparent differences between classical gravity and quantum field theory, the geometric structure of the two bear a surprising resemblance. Employing the Einstein–Cartan formalism, it is possible to describe gravitational interactions by the means of a local gauge group. Hence, Einstein–Cartan gravity can be regarded as an instance of Poincaré gauge theory [55, 56], which incorporates both the curvature and the torsion of the manifold. The curvature and torsion tensors are the field strengths of the non-abelian spin-connection $\omega_\nu{}^\mu$ and the abelian tetrad gauge field $e_\mu{}^a$, respectively. The spin connection $\omega_\nu{}^\mu$ pertains to rotations and boosts, and the tetrad field $e_\mu{}^a$ to translations. Together they yield the covariant

derivative $\partial_\mu \longrightarrow \mathcal{D}_\mu = \partial_\mu + e_\mu{}^a P_a + \omega_\mu{}^{ab} M_{ab}$ of the theory. Evidently, this takes on a similar form to that of SU(N) Yang–Mills theory, but with gauge group generators $P_a$ and $M_{ab}$, which span the group of translations and Lorentz transformations.

The main objective of this work is to construct a non-abelian gravitational analog of the Aharonov–Bohm flux tube to which we wish to couple spinors. In order to achieve this, it is convenient to formulate the gravitational theory in a local coordinate basis where the Poincaré group $\text{ISO}^+(1,2)$ of space-time symmetries plays the role of a gauge group. The aforementioned spin-connection is an object which provides a connection between the local frames on the manifold. The spin-connection is a Lie-algebra valued 1-form of the Poincaré group, meaning that we can regard it as a non-abelian gravitational gauge field. To obtain the spin-connection, the effective metric simulated by the condensate is required.

### 3.1 Einstein–Hilbert action and the acoustic metric

Einstein field equations of general relativity can be derived from the Einstein–Hilbert action

$$S_{\text{EH}} = \frac{c^4}{16\pi G} \int dx^4 R \sqrt{-g}\,, \tag{2}$$

where $R$ is the Ricci scalar, $g$ is the determinant of the space-time metric $g_{\mu\nu}$, $c$ is the speed of light and $G$ is Newton's gravitational constant. In a (2+1)-dimensional superfluid an emergent analog gravity picture can be cast into a similar form [15] with an additional term accounting for the quantum pressure

$$S_{\text{SEH}} = \frac{c_s^4}{16\pi} \int dx^3 (\phi R - \frac{\gamma}{\phi^{3/2}} \nabla^2 \phi)\,, \tag{3}$$

where $c_s$ is the speed of sound in the superfluid, which is the analog of the speed of light, $\phi$ can be regarded as an analog dark matter scalar field to which the curvature is coupled, and $\gamma$ is a combination of natural constants. The distance in the superfluid universe is governed by the so called acoustic metric [4]

$$g_{\mu\nu}(r) = \Omega^2(r) \left( \begin{array}{c|c} -(c_s^2 - v_s^2) & -v_i \\ \hline -v_j & \delta_{ij} \end{array} \right)\,, \tag{4}$$

where $\Omega(r)$ is a conformal factor proportional to the condensate density $|\Psi(r)|^2$, and $v_i$ are the components of the superfluid velocity $\vec{v}_s$. We temporarily consider metrics with rotational symmetry with no vortices present in the proximity such that the torsion vanishes in the long-distance limit. Subject to these conditions, the resulting invariant line-element in polar coordinates may be expressed as

$$ds^2 = \Omega^2(r)(-c_s^2 dt^2 + dr^2 + r^2 d\theta^2)\,. \tag{5}$$

Note that in this regime the metric is conformal to that of Minkowski space where the conformal factor $\Omega(r)$ determines the distance. In order to gauge the gravitational theory, we set up a flat coordinate system at each point in the analog space-time. This can be achieved within the tetrad formalism by introducing the fields $e_\mu{}^a$, defined such that $\eta_{ab} = e_a{}^\mu e_b{}^\nu g_{\mu\nu}$, where $\eta_{ab}$ represents the flat tangent-space metric. We use latin indices to denote local coordinate components and greek indices for the global ones. In light of Eq. (5) the tangent-space basis vectors $e^a = e^a{}_\mu dx^\mu$ can be defined as

$$e^t = \Omega(r)c_s dt, \quad e^r = \Omega(r)dr, \text{ and } e^\theta = \Omega(r)r d\theta\,. \tag{6}$$

The set of equations (6) may further be used to compute the spin-connection via Cartan's structure equation

$$T^a = de^a + \omega^a{}_b \wedge e^b, \tag{7}$$

where $T^a$ denotes the torsion, which vanishes in absence of vortices, and $\omega^a{}_b$ represents the components of the spin-connection. The torsion of the manifold can be regarded as the field strength tensor associated with the abelian tetrad gauge fields $e_\mu{}^a$. Equation (7) may be solved using the anti-symmetry of the wedge product and by noting that the exterior derivative of any infinitesimal element must vanish (Poincaré's lemma), that is $d(da) = 0$ for all $a$. To simplify the calculation, the relation $\omega^a{}_b = \eta^{ac}\omega_{cb}$ may be used to establish that $\omega^a{}_b$ must be symmetric in the time components and anti-symmetric in the spatial components. This is because Lorentz invariance of $\eta_{\mu\nu}$ implies that $\omega_{\mu\nu} - \omega_{\nu\mu} = 0$, that is, it must be anti-symmetric in all components, and consequently, $\omega^\mu{}_\nu$ must be anti-symmetric only in the spatial components. With these considerations, we obtain the 1-form solutions

$$\omega^t{}_r(r) = \omega^r{}_t(r) = c_s \frac{\partial_r \Omega(r)}{\Omega(r)} dt \tag{8}$$

and

$$\omega^r{}_\theta(r) = -\omega^\theta{}_r(r) = \left( r\frac{\partial_r \Omega(r)}{\Omega(r)} + 1 \right) d\theta, \tag{9}$$

and all other components vanish. Note the explicit appearance of the density gradient in the non-zero components of the spin-connection which accounts for the effective gravity. In fact, the factor

$$v_{\nabla\Psi} = \frac{\partial_r \Omega(r)}{\Omega(r)} \tag{10}$$

corresponds to a velocity field $v_{\nabla\Psi}$ induced by the quantum pressure. In total, there are two sources for a fluid velocity: one from the conventional phase gradient $v_{\nabla\theta}$ and one coming from the density gradient. The former give rise to the well-known U(1) phase winding and the latter, as we shall see, results in an arbitrary rotation. This is due to the fact that while the phase winding is quantized, there is no such restriction imposed on the velocity due to the quantum pressure, thus enabling arbitrary rotations of the quasi-particle spinor. The gravitational phase can be understood from the point of view of frame transformations. A description of the physics in the quasi-particle's frame of reference can be obtained by adding an angular momentum operator term $W_{\nabla\Psi}L_z$, where $W_{\nabla\Psi} = \frac{1}{r}v_{\nabla\Psi}$ is the orbital velocity. Transforming to the rest frame of the quasi-particle can thus be considered equivalent to introducing an artificial gauge field. The term accounting for the kinetic energy is thereby taking on a similar form to that of an electron minimally coupled to a magnetic field. Hence, the gravitational interaction thus looks essentially the same as an electromagnetic interaction where the flux is parametrized by $v_{\nabla\Psi}$. In the BdG description, to be explicit, the $(u, v)^T$ spinor thus evolves due to gravity according to

$$\begin{pmatrix} u \\ v \end{pmatrix} \longrightarrow \exp iH_G t \begin{pmatrix} u \\ v \end{pmatrix}, \tag{11}$$

where

$$H_G = \begin{pmatrix} (\nabla - v_{\nabla\Psi})^2 & 0 \\ 0 & -(\nabla - v_{\nabla\Psi})^2 \end{pmatrix} \tag{12}$$

is the gravitational part of the BdG Hamiltonian. The result of bringing a quasi-particle spinor around a generic topological defect is therefore an arbitrary Pauli-$Z$ rotation diag($e^{i\theta}, e^{-i\theta}$) in contrast to that of a conventional vortex diag($e^{in2\pi}, e^{-in2\pi}$) whose action is trivial. Recall that our goal is to reproduce an Aharonov–Bohm flux tube, that is, to construct a space-time

with vanishing curvature everywhere except inside a finite disk $\Sigma$ of radius $r_\circ$. The curvature 2-form of the spin-connection is

$$R^a{}_b = d\omega^a{}_b + \omega^a{}_c \wedge \omega^c{}_b \,. \tag{13}$$

Again, we highlight the analogy with non-abelian Yang–Mills theory where the field strength is given by $F_{\mu\nu} = \partial_\mu A_\nu - \partial_\nu A_\mu - i[A_\mu, A_\nu]$.

# 4 A topological theory

## 4.1 Pauli-$Z$ from gravitational holonomy

For now, let us focus on the spatial components of the connection defined in Eq. (9). For this connection to be flat, Eq. (9) must be equal to an arbitrary real constant $C$, so that $R^a{}_b$ vanishes. That is, we wish to solve the following differential equation for $\Omega(r)$

$$\omega^\theta{}_r = r\frac{\partial_r \Omega(r)}{\Omega(r)} + 1 = C \,, \tag{14}$$

which has the solution

$$\Omega(r > r_\circ) = Ar^{C-1} \,, \tag{15}$$

where $A$ is an arbitrary real constant. The solution takes on this particular form since the function either vanishes or diverges at $r = 0$ for $C > 1$ and $C < 1$, respectively, with both cases corresponding to a topological defect. Interestingly, if we insert this density profile into the gravitational term in Eq. (1), we obtain

$$|\Psi_{\mathrm{G}}(r)|^2 = \left(\frac{C-1}{2r}\right)^2 \,. \tag{16}$$

This term is, up to a constant, identical to the electromagnetic phase gradient term

$$|\Psi_{\mathrm{EM}}(r)|^2 = \left(\frac{n}{r}\right)^2 \,, \tag{17}$$

where $n$ is the winding number of the vortex. Consequently, this leads to the following proposition:

**Proposition** (P.G.–Q.P. equivalence)
*Let $\mathcal{H} = \mathcal{H}_{P.G.} + \mathcal{H}_{Q.P.}$ be the Hamiltonian accounting for the phase gradient and quantum pressure. Then, if we choose a scalar field density $|\Psi(r)|^2 = Ar^{C-1}$, where $A$ is an arbitrary constant, the two terms are equivalent if we pick $C$ such that $C - 1 = 2n$, where $n$ is the winding number of the vortex with Hamiltonian $\mathcal{H}_{P.G.}$.*

Moreover, since the vortex phase is $2\pi n$, one can directly infer from the Hamiltonian that the gravitational phase is determined by the constant $C$. Now, for a flat exterior, we may introduce a variable density in the interior of the disk such that $\omega^a{}_b(r)$ is smooth on the boundary $\partial\Sigma$. An example of such a function is

$$\omega^a{}_b(r \leq R) = \frac{C}{R^4}r^4 - \frac{2C}{R^2}r^2 \,, \tag{18}$$

where the coefficients have been chosen such that $\omega^a{}_b(r)$ is smooth on $\partial\Sigma$. This choice leads to the following differential equation in the interior

$$r\frac{\partial_r \Omega(r)}{\Omega(r)} + 1 = \frac{C}{R^4}r^4 - \frac{2C}{R^2}r^2 \,, \tag{19}$$

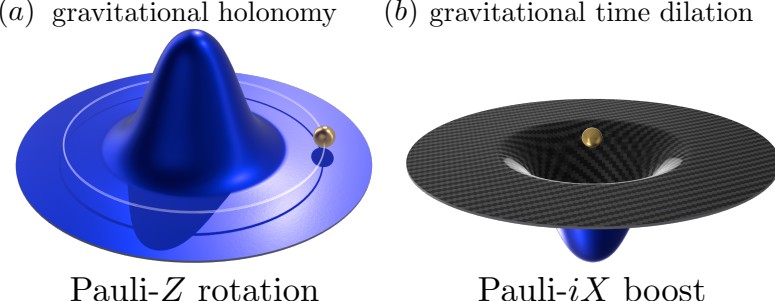

$(a)$ gravitational holonomy    $(b)$ gravitational time dilation

Pauli-$Z$ rotation    Pauli-$iX$ boost

Figure 1: (a) Particle (sphere) encircling a curvature bulge (or dent) may be used for realizing a Pauli-$Z$ gate. (b) Particle (sphere) spending time in a curvature dent (or bulge) may be used for realizing a Pauli-$iX$ gate.

with a solution

$$\Omega(r \leq r_\circ) = B r^{-1} e^{\frac{1}{4}\left(\frac{C}{R^4} r^4 - \frac{2C}{R^2} r^2\right)}. \tag{20}$$

Henceforth, let us set $A = B = 1$ since the resulting curvature is independent of this choice. We let $C$ be a free control parameter that can be adjusted to acquire a desired phase. With these considerations, the geometric phase may be calculated. Given a connection 1-form $\omega$, the geometric phase acquired by a wave-function $\Psi(r, t)$ under encirclement is

$$\Psi(r, t) \longrightarrow \mathcal{P} e^{i \frac{g}{c_s \hbar} \int_\gamma \omega} \Psi(r, t), \tag{21}$$

where $g$ is a coupling constant and $\mathcal{P}$ denotes path-ordering. To highlight the topological nature of the configuration, we may apply the generalized Stokes' theorem which states that the contour integral of a connection 1-form $\omega$ along a closed boundary $\partial S$ is equal to the surface integral of its curvature 2-form over the surface $S$. A straight forward computation yields

$$\int_{\partial S} \omega = \int_\Sigma (d\omega + \omega \wedge \omega) = 2\pi C, \tag{22}$$

since the only contribution comes from the non-vanishing curvature in the interior of the disk $\Sigma \subset S$. This means that the shape of the path has no bearing on the outcome and the phase is protected by topological equivalence. Note that the phase computed in Eq. (22) corresponds to a rotation in the $xy$-plane, see Fig. 1(a), since it was computed from $\omega^r{}_\theta$ component which yields a Pauli-$Z$ rotation given a spinorial representation. In conclusion, the topological non-abelian phase acquired by a quasi-particle, owing to holonomy, is determined by elements in the group

$$\mathrm{Hol}(\omega = C) = e^{i \frac{2\pi g}{c_s \hbar} C \sigma_3}, \tag{23}$$

where $\sigma_3$ is the Pauli-$Z$ matrix. Since $C$ is a parameter which determines the density profile that can be adjusted in the laboratory, an arbitrary topological phase can be implemented in this way. Note that the disk $\Sigma$ is a true topological defect of a gravitational character with topological charge $C$, akin to a radially stretched cosmic string in two spatial dimensions, where the curvature makes a discontinuous jump to a non-zero value at the boundary $\partial \Sigma$.

## 4.2 Pauli-$iX$ from time dilation

Let us next consider the temporal part of the connection. Given the density profile in the exterior defined by Eq. (15), $\omega^t{}_r(r)$ takes the form $\omega^t{}_r(r) = c_s(C - 1)r^{-1} dt$ which in the

spinorial representation results in a transformation $e^{K(t)}$, where

$$K(t) = c_s(C-1)r^{-1}\int_{\Delta t} dt\sigma_1 \tag{24}$$

is a Lorentz boost. Thus the state will accumulate an additional Pauli-$iX$ transformation during the time $\Delta t$ it spends in the region with a conformal factor given by Eq. (15). However, since this contribution is always proportional to $r^{-1}$, it can be made arbitrarily small by considering long distance scales. This transformation could also be utilized to do computation by choosing a density profile such that the region in the vicinity of the particle is curved, while the curvature vanishes everywhere else. Thus, by holding $\Omega(r)$ constant such that $\omega^t{}_r(r) = 0$ everywhere but in a small region in which the quasi-particle resides, an additional transformation can be implemented. Ideally we would like to do this in a path-independent way as well. This may be accomplished by letting

$$\Omega(r) = De^{\frac{C}{c_s}r}, \tag{25}$$

since this choice of $\Omega(r)$ yields a constant $\omega^t{}_r(r) = C$ in the neighbourhood of the quasi-particle. The arbitrary constant $D$ has no influence on the transformation so we may set $D = 1$ as well. Defining the density landscape in this way, the quasi-particle state will transform due to analog time dilation according to the group of Lorentz boosts

$$\text{Boost}(\omega = C) = e^{\frac{g}{c_s\hbar}\int_{\Delta t} Cdt\sigma_1}, \tag{26}$$

parametrized by $C$. Note that $\omega^r{}_\theta(r) \propto r$ for this choice of $\Omega(r)$, which vanishes in the neighbourhood of the quasi-particle. We wish to emphasise that this transformation is not topological in the same sense as the holonomy gate. While the spatial deformations of the world line trajectories leave the result invariant, the outcome will depend on the duration of the interaction. Or to put it differently: the transformation accumulated is independent of the path taken between the initial and final time slice, but it will depend on the distance between the two slices. It is therefore imperative to switch gravity off at the right instance in order to avoid errors.

Let us next outline a few pertinent remarks regarding this particular analog universe. Given the metric defined by Eq. (5), and the density profile defined by the conformal factor in Eq. (25), the Lorentz invariant line element takes on the form

$$ds^2 = e^{2\frac{C}{c_s}r}(-c_s^2 dt^2 + dr^2 + r^2 d\theta^2). \tag{27}$$

This analog space-time is thus identical to the so called Rindler metric [40], which describes the coordinates experienced by an observer in a non-inertial frame of reference, subjected to a constant acceleration. The Rindler metric was one of the corner stones of the calculation carried out by Unruh in 1976 [57], which predicted that the mode expansion observed in one frame of reference is different from that observed in another, given a non-zero relative constant acceleration between the two. Hence, the notion of a state, and thus of particle number, is ambiguous since it is frame dependent. In particular, the vacuum state observed by an inertial observer is not a vacuum for an accelerated one. The accelerated observer will see a thermal heat bath of spin-0 particles governed by the Bose–Einstein distribution [51] and the mode expansions in the two frames are related via a Bogoliubov transformation [52]. The phenomenon described here is often referred to as the Unruh effect which certainly is one of the most striking phenomena predicted by quantum field theory in a curved space-time [58–60].

# 5  On computational universality

We have shown that unitary transformations due to gravity can be harnessed in two distinct ways. It is thus natural to pose the question whether these transformations form a universal gate set for the purposes of quantum information processing. In order for universal single-qubit quantum computation to be possible, the unitary transformations must be able to map any point on the Bloch-sphere to any other point on it, in a topologically dense way. It is clear that any rotation about the $z$-axis can be achieved by means of the holonomy gate. However it is not obvious that the boost will provide the additional transformations required to achieve this. In fact, it turns out that the boost, just as the holonomy, is only capable of mapping a point on one hemisphere to other points on the same hemispehere. That is, the gravitational gate set is only universal on one hemisphere at a time, and will not allow crossing the equator. For instance, if we start in a state $|1\rangle = (1,0)^T$ pointing at the north pole, the boost will cause the time-evolving state to continuously rotate towards the equator, but it will never reach it. Instead, the result is equivalent to the action with a Hadamard gate. Mathematically we can describe this process as

$$\lim_{t\to\infty} e^{\int_0^t Cdt\sigma_1} |1\rangle = \frac{1}{\sqrt{2}} |0\rangle + \frac{1}{\sqrt{2}} |1\rangle \, . \tag{28}$$

The same is true if we start in the state $|0\rangle = (0,1)^T$. What happens when the spinor is subjected to gravity is that the Lorentz boost will transform the coefficients as a function of the rapidity $\alpha = Ct$, causing the spinor amplitudes to mix. Consequently, if we equip the gravitational gate set with an additional operation $\mathcal{O}$ that is nudging the Bloch vector across the equator, the resulting single qubit gate set becomes universal. Note that the operation $\mathcal{O}$ need not be topological since it does not matter where on the other hemisphere the state vector lands. As long as the equator can be crossed, every point on the sphere can be reached since the gravitational gate set is universal on each of the hemispheres separately. We may therefore conclude that the gate set $\{\text{Hol}(\omega), \text{Boost}(\omega), \mathcal{O}\}$ is capable of universal single-qubit quantum computation given that the density profiles are chosen accordingly. Whether such gravity-only operations can be extended to multi-qubit systems remains an open question.

# 6  Conclusions

We have shown that analog gravitational interactions in a scalar field theory can be implemented topologically by carefully choosing the density profile. For the analog universes studied in this work there are two distinct effects, owing to holonomy and time dilation, from which unitary quantum gates can be harnessed. These interactions are mediated via the spin-connection, which serves as an analog gravitational gauge field. We emphasise that our proposal is generic and can be applied to analog gravity systems independent of their system specific details. As discussed, one may exploit only one of these effects to supplement a pre-existing non-universal quantum computation model, e.g. MZM models or quantum doubles, to achieve universality. One may, for instance, consider a holonomy gate and pick the constant $C$ to be irrational $C \in \mathbb{R} \setminus \mathbb{Q}$. With this choice of $C$, the resulting gate must be of infinite order implying that any single qubit braid set with this gate added will span the entire Bloch-sphere, since two non-commuting elements of infinite order is sufficient for this. Also, since the Ising anyon braid group, which is the braid group pertaining to the MZMs, generates the Clifford group, universality can be attained by adding a $\pi/8$-gate, which could be implemented by the $C = \frac{1}{8}$ gravitational holonomy gate.

Regarding potential future work, a natural direction would be to capitalize further on the

gravitational gate set in an attempt to develop a gravity-only platform for quantum computation. Given such a gate set, universal quantum computation could be achieved gravitationally with no braiding required. In fact, true non-abelian anyons are not even required since the gravitational effects are present independent of whether the quasi-particle spinors span a degenerate ground state. All quasi-particle spinors couple to metric studied here in a space-time endowed with an acoustic metric. Analogue gravity may therefore offer a pathway to circumvent the issue of realizing true non-abelian anyons in experiments. A potential candidate that could be more experimentally tractable is the kelvon quasi-particle [61, 62] in scalar Bose–Einstein condensates. Kelvon quasi-particles correspond to linear perturbations of the fluid and are localized within the cores of quantized vortices. Thus, a kelvon is inseparable from its host vortex meaning that it can be moved around and pinned by simply trapping the vortex in an external potential. Controlling the vortex in this way enables the implementation of the gravitational gates on the kelvon spinor. The exact architecture of such a platform, including the implementation of the additional operation $\mathcal{O}$, will be left for future work together with the open question of how to implement two-qubit controlled gates gravitationally.

# Acknowledgements

We are grateful to Chris Vale for useful discussions.

**Funding information** We acknowledge financial support from the Australian Research Council via the Future Fellowship Project FT180100020.

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
