# Peer review of "Topological quantum computation using analog gravitational holonomy and time dilation"

_SciPost Physics Core, doi:SciPost Phys. Core 6, 005 (2023)_

## Round 1 · Referee Report · Anonymous (Referee 1) · 2022-8-1

Strengths

Original and interesting idea for mechanism to realize quantum gates exploiting analog gravity as it arises in superfluid condensates. Explicit construction of field configuration for the required non-abelian energy-momentum flux tubes.

Weaknesses

Mechanism for carrying quasi-particle excitations around flux tube is left open; proposed scheme is for 1-qubit gates only and needs to be supplemented by 2-qubit gates.

Report

This paper proposes to exploit the ``analog gravity” arising in non-uniform superfluid condensates for a mechanism to realize quantum gates on quasi-particle excitations. Key ingredient is an Aharanov-Bohm effect for a carefully designed non-abelian energy-momentum flux tube. The proposed scenario is original and interesting but, at this stage, a bit unbalanced. While the gravitational field configurations are worked out in mathematical detail, the mechanisms for creating such field configurations and carrying quasi-particle excitations around a flux tube are left open.

Requested changes

I would like to invite the authors to address and clarify a number of points.

1. It would be helpful to this reader to give a bit more detail on how the quasiparticles (arising as spinors in a BdG formalism) transform under gravitational transformations, backing up expressions such as eq. (21), (22).
2. The ``pertinent remarks” on the Unruh effect (end of section 4) seem unrelated to the gravitational holonomy gates. Is there a connection, or a physical effect that’s relevant in this context?
3. In section 4.2, it is unclear to this reader in what sense the mechanism for the Pauli-iX boost is topological, as the effect seems to depend on a time $\Delta t$ which is not a discrete quantity.
4. I find it confusing that section 5 uses the term ``universality” in the sense of being dense on the 1-qubit Bloch sphere, leaving aside the 2-qubit gates.
5. In section 6 it is suggested that gravitational holonomy gates supplement a quantum computational scheme based on MZM or quantum doubles. Can this be made more concrete – for example could the analog gravity scenario be realized in a $p_x+ip_y$ superfluid that supports MZM?
6. In section 6 it is suggested that a gravity-only platform for TQC can be developed, for example by employing the kelvon quasi-particles. It is unclear to me how 2-qubit gates can be realized in such a scenario?

  • validity: high
  • significance: high
  • originality: high
  • clarity: good
  • formatting: excellent
  • grammar: excellent

Author:  Emil Génetay Johansen  on 2022-09-22  [id 2842]

(in reply to Report 1 on 2022-08-01)
Category:
answer to question

Dear referee,

Please see the file attached containing our detailed response to the report.

Best regards,
Emil Génetay Johansen and Tapio Simula

Attachment:

Response_letter.pdf

---

## Round 2 · Referee Report · Anonymous (Referee 1) · 2022-10-10

Report

The authors have carefully addressed and clarified the points I made in my first report, and made some adjustments to the manuscript accordingly. While many open ends remain, I reiterate that the proposed scenario is original and interesting, and I'm happy to recommend the manuscript for publication in SciPost Physics Core.

---

## Round 2 · Author Response

Please find below (under list of changes) the questions asked by the referee and replies containing information about where in the manuscript the questions are addressed. Also a document in which we respond to the referee's report is shared on the report page.

---

## Round 2 · List of Changes

Requested changes

  1. It would be helpful to this reader to give a bit more detail on how the quasiparticles (arising as spinors in a BdG formalism) transform under gravitational transformations, backing up expressions such as eq. (21), (22).

The gravitational interaction can be described in terms of an ``artificial" gauge field. The gravitational connection 1-forms depend on the velocity in Eq.(10), which arises due to a (probability) density gradient and the source of the gravitational transformation is the quantum pressure term. Transforming to a rotating frame of reference is generically achieved by adding an angular momentum operator multiplied by an angular velocity to the Hamiltonian, which can be expressed in terms of the velocity due to gravity in Eq.(10). The term accounting for the kinetic energy is thereby taking on a similar form to that of an electron minimally coupled to a magnetic field. The gravitational interaction thus looks essentially the same as an electromagnetic interaction on a charged particle. We have added clarifying sentences and expressions in the revised manuscript. See Eq.(11), (12) and the surrounding text.

  1. The pertinent remarks” on the Unruh effect (end of section 4) seem unrelated to the gravitational holonomy gates. Is there a connection, or a physical effect that’s relevant in this context?

The referee is correct. The analogy to the Unruh effect is not relevant for the holonomy, only for the Lorentz boost. The time components of the spin connection (first row and first column) belong to the boost part of the Poincaré group while all other components (the spatial ones) belong to the rotational part. Thus, gravity is implementing a boost on the spinor when the time components are non-zero resulting in a transfer in amplitude between the spinor components. The rotational part on the other hand is leaving the distribution of the amplitude invariant. The analogy to the Unruh effect has therefore been described in the manuscript for the boost gate, not for the holonomy gate.

  1. In section 4.2, it is unclear to this reader in what sense the mechanism for the Pauli-iX boost is topological, as the effect seems to depend on a time Δt which is not a discrete quantity.

We are grateful to the referee for drawing our attention to this point. Indeed, the boost gate is not topologically protected in the conventional sense because its action depends explicitly on the continuous time variable. Prompted by the referee’s observation, we have sharpened our wordings on this in the manuscript. However, it is true that space-time trajectory between two time slices has no bearing on the outcome, which only depends on the distance between the two slices. We have clarified this in section 4.2.

  1. I find it confusing that section 5 uses the term universality” in the sense of being dense on the 1-qubit Bloch sphere, leaving aside the 2-qubit gates.

We thank the referee for highlighting this. Indeed, the gate set described is providing a dense cover in SU(2) and as such is only universal on a 1-qubit level (clarification on this is added in section 5). However, given a 4-spinor the gravitational gate should act as a spin-3/2 representation of SU(2) thus implying that the gravitational gates should be applicable to qudit systems as well.

  1. In section 6 it is suggested that gravitational holonomy gates supplement a quantum computational scheme based on MZM or quantum doubles. Can this be made more concrete – for example could the analog gravity scenario be realized in a px+ipy superfluid that supports MZM?

Our result is extremely general the only assumptions being that the underlying fluid can be characterised by an acoustic metric, and that its excitations carry a spin. As such, we do indeed expect our results to be applicable to both MZM vortices in topological superconductors, e.g. Nature 606, 890 (2022), and to non-abelian vortex anyons in spinor BECs, e.g. Phys. Rev. Lett. 123,140404 (2019).

  1. In section 6 it is suggested that a gravity-only platform for TQC can be developed, for example by employing the kelvon quasi-particles. It is unclear to me how 2-qubit gates can be realized in such a scenario?

The referee is correct in that 2-qubit gates are required in order to realize arbitrary quantum algorithms. In order for the gravitational gates to act as a spin-3/2 representation on the kelvon states one would need to find means to entangle two kelvons gravitationally. At this stage it is not clear to us if this can be achieved without developing an analogue model of quantum gravity. In the manuscript, we do not claim that it can, and merely suggest that it would be a natural direction to explore in the future.

---

## Editorial Decision

published